# DEDNet: Offshore Eddy Detection and Location with HF Radar by Deep Learning

**DOI:** 10.3390/s21010126

**Published:** 2020-12-28

**Authors:** Fangyuan Liu, Hao Zhou, Biyang Wen

**Affiliations:** School of Electronic Information, Wuhan University, Wuhan 430072, China; liufy@whu.edu.cn (F.L.); bywen@whu.edu.cn (B.W.)

**Keywords:** eddy detection, pixel-wise segmentation, high frequency radar, sea current

## Abstract

Oceanic eddy is a common natural phenomenon that has large influence on human activities, and the measurement and detection of offshore eddies are significant for oceanographic research. The previous classical detecting methods, such as the Okubo–Weiss algorithm (OW), vector geometry algorithm (VG), and winding angles algorithm (WA), not only depend on expert’s experiences to set an accurate threshold, but also need heavy calculations for large detection regions. Differently from the previous works, this paper proposes a deep eddy detection neural network with pixel segmentation skeleton on high frequency radar (HFR) data, namely, the deep eddy detection network (DEDNet). An offshore eddy detection dataset is firstly constructed, which has origins from the sea surface current data measured by two HFR systems on the South China Sea. Then, a spatial globally optimum and strong detail-distinguishing pixel segmentation network is presented to automatically detect and localize offshore eddies in a flow chart. An eddy detection network based on fully convolutional networks (FCN) is also presented for comparison with DEDNet. Experimental results show that DEDNet performs better than the FCN-based eddy detection network and is competitive with the classical statistics-based methods.

## 1. Introduction

Oceanic eddy is a common phenomenon of seawater flow, which plays an important role in transporting energy and matter. Usually, an eddy is defined as a lateral current structure with velocity vectors rotating around a center in a clockwise or counterclockwise direction [1]. In oceanography, mesoscale eddies normally have sizes of 1–100 km and can survive up from 1 h to 1 year [2], and transports mass, heat, momentum, marine organisms, and biogeochemical along certain trajectories. Meanwhile, oceanic mesoscale eddies also have an interaction effective on clouds, winds, and rainfall, which indirectly influence the atmospheric dynamic environment [3,4]. Therefore, the detection and tracking of mesoscale eddies have important implications in the field of marine environment and atmospheric environment. Particularly, submesoscale eddies, which are between one to ten km in diameter and survive for periods of hours to days, play an important role as a link between the mesoscale eddies and the turbulent currents in the mass and energy transfer and thus are a key component of the global heat budget [5].

As for eddy detection, the main eddy characteristics include the center’s position, radius, and boundary [6]. To extract the characteristic information, different equipment can be used to collect data. The popular observation equipment includes in-situ sensors, satellite altimeters [7], and high-frequency radars (HFRs) [8], each providing valuable observations with different spatial resolutions. The in-situ sensors have an extremely high resolution but are difficult and inefficient to track the spatial variation in a large area. Satellite altimeters are capable of large-area observations of surface currents, but the samples are coarse-grained, since most marine satellites work with a low spatial resolution (larger than 0.25° × 0.25°) and a long revisit period (far more than 1 h). HFRs are capable of accurate offshore eddy detection with a relatively higher spatial resolution 0.05° × 0.05°) and a relatively shorter revisit period (20 min). HFRs are particularly suitable for observation of small-area fine-grained eddy detection and therefore owe an irreplaceable value on submesoscale eddy detection. The growth of the worldwide HFR network also promotes the global and regional current model for subsequent studies on ocean dynamics. In this paper, submesoscale eddy detection from HFR current data with a deep learning network is addressed.

The widely used eddy detection methods include statistics-based methods [9] and deep learning (DL) methods [10]. The statistics-based methods can further be divided into expert-based, physics-based, and geometry-based methods. The expert-based methods use experts’ experiences of vision, which is the most reasonable but difficult to handle large-scale task of eddy detection. The physics-based methods detect eddies by setting thresholds on physical quantities [11], such as current speed, vorticity, pressure, and helicity. Geometry-based methods detect eddies by computing geometric criteria in the candidate regions [12,13]. Usually, in physics-based methods, the shape indices of the streamlines are calculated to help determine whether there exist revolving streamlines in the region. The outermost closed contour of the streamlines is calculated and regarded as the eddy boundary, and the zero-speed position within the eddy boundary is regarded as the eddy center. Differently from the physics-based methods, geometry-based methods extract the eddy’s shallow features through one transformation and then determine whether an eddy exists in the observed region according to these shallow features.

With the rapid development of DL, a growing number of sensor systems adopt a deep learning model for object localization and tracking. The DL method offers another approach for eddy detection, which is able to extract eddies’ deep features from multiple layers, recombine features across different layers, and simultaneously identify the centers and boundaries [14,15]. According to the properties of tasks, the DL network can be divided into two branches: the objective detection baseline [16] and the semantic segmentation baseline [17]. In the objective detection baseline, the common structure is an objective detection neural network added with the vector geometry (VG) algorithm [18]. The objective detection neural network is responsible for determining the possible eddy regions, reducing the candidate regions’ area, and decreasing the overlapped area of different candidate regions. The VG algorithm is responsible for determining the eddy centers and calculating the boundaries. Actually, the performance of the object detection baseline mainly depends on the VG’s performance, while the neural network just plays a role in localizing the possible regions. In the semantic segmentation baseline, the eddy detection task is regarded as a pixel-wise classification, which divides the pixels of the flow chart into multiple categories, such as cyclones, anticyclones, and background. The whole pipeline [19] only covers the neural network, which reduces the influence from the statistical method. Our work follows the semantic segmentation baseline to implement pixel-wise classification from the HFR observation data.

In this paper, a deep eddy detection neural network for HFR observation data, namely DEDNet, is proposed to automatically detect eddies and report their centers and boundaries. Firstly, in the pyramid scene parsing network (PSPNet)-like skeleton, we present an efficient deep detection network for HFR observation data, which considers both the global regional feature and the detailed geometry feature of an eddy. Secondly, in the FCN-based skeleton, we present a portable eddy detection network for performance comparison. Thirdly, for repeatable comparison experiments, we collect and construct eddy detection datasets by the real HFR data from the South China Sea.

The remainder of this paper is expressed as follows. Section 2 introduces the background of eddy detection. Section 3 shows data preparation for eddy detection. Section 4 describes the architecture and the loss metric of DEDNet. Section 5 demonstrates the performance assessment and result analysis. Section 6 gives the conclusion.

## 2. Related Work

The development of eddy detection depends on two aspects: the observation equipment and the detection algorithm. The observation equipment collects as much data as possible and provides a strong support to the detection algorithm. The detection algorithm extracts useful information from the collected data and implements the eddy detection tasks. Therefore, both aspects are important.

As for the observation equipment, the development involves in-situ equipment and remote sensing tools such as satellite altimeters and HFR. The in-situ observation is an inefficient means of eddy detection, whether using an anchored or drifting platform. For example, considering that the eddy is a random and hardly predicted phenomenon, the drifting platform cannot completely cover all possible eddy regions and may only collect the information of eddies by coincidence. Satellite altimeter can provide statistical information of surface eddies by detecting sea surface height anomalies (SSHA) [20], sea level anomalies (SLA) [21], or sea surface temperature (SST) [22], which is then applied to compute the lifetime, eddy radius, spatial distribution, trajectory, and vorticity of each eddy. The studies on oceanic eddies develop rapidly due to the increasing open datasets for oceanographic researches, e.g., the Copernicus marine environment monitoring service (CMEMS) datasets [7]. However, only a small number of satellites have sufficient resolution to provide accurate eddy measurements, and satellite-altimeter-based eddy studies tend to focus on specific regions where the observation data are sufficient for investigation, for example, the Mediterranean and Australian Coral Sea region. Other regions achieve much less attention. So far, HFR is the most efficient equipment for eddy detection in specific oceanic regions such as the offshore seas, bays, and regions around rigs. HFR has the characteristics of high spatial–temporal resolution and low configuration cost. With the growing number of HFR all over the world, it becomes an irreplaceable tool and plays a more and more important role in sea state monitoring.

As for the detection algorithm, the development experiences the expert’s visual detection, physics-based algorithms, geometry-based algorithms, and deep learning methods. Depending on expert’s experience, visual detection manually detects eddies and distinguishes their boundaries. It is an accurate decision approach, but it is laborious and time-consuming, which makes it impossible to simultaneously detect large number of eddies in a large region. Physics-based algorithms detect eddies by measuring the vorticity, pressure, helicity, and gradient quantities. An experience-based threshold is assigned to distinguish the eddy regions from the no-eddy regions. One classical physics-based algorithm is the Okubo–Weiss algorithm [23], which defines three direct indices, including shearing deformation rate, straining deformation rate, and vorticity, to determine whether the region contains eddies. Considering that the physics-based algorithms rely on the choice of the threshold, it may be easily affected by experientialism and human interference, which is not the best choice for repetitive eddy detection tasks. Geometry-based algorithms apply a geometric standard to identify eddies. The representative algorithms are the vector-geometry (VG) [18,24] and winding-angle (WA) [25] algorithms, which directly calculate the swirling pattern around the center. The VG algorithm applies four constraint conditions to decide the eddy center and takes the streamlines’ outermost closed contour as the eddy boundary, while the WA algorithm clusters the streamline centers and takes the center of the centers in the same cluster as the eddy center, and achieves the boundary by fitting to the streamlines in the same cluster. The VG algorithm uses iterative searches to achieve accurate streamlines, which leads to a heavy burden in computation [26].

Differently from the traditional eddy detection methods, the DL method regards eddy detection as a specific computer vision (CV) task. General neural networks have been tried to implement eddy detection from different viewpoints. The mainstream viewpoint can be divided into two branches: objective detection tasks and semantic segmentation tasks.

In the object detection branch, the common sequential structure is an objective detection neural network to locate the eddy’s position followed by the VG algorithm to determine the eddy’s center and boundary. Ocean Eddy Identification Neural Networks (OEDNet) was constructed for automatic identification and positioning of mesoscale eddies [27], the skeleton of which includes a RetinaNet, a deep residual network, and a feature pyramid network. It uses multiple SLA data to search for mesoscale eddies with small samples and in complex regions. Xu and Cheng proposed an artificial intelligence algorithm for eddy detection based on PSPNet and the VG algorithm [28]. Limited by the VG algorithm, the accuracy remains on a similar level as the geometry-based algorithm. Du and Wang proposed an eddy identification and tracking framework mainly based on feature learning with convolutional neural network and using the SLA data of Australia [29]. As a conclusion, the performance of the objective detection branch is limited by the VG algorithm, since it relies on the VG algorithm to determine the eddy’s center and boundary.

In the semantic segmentation branch, EddyNet [30] is a neural network based on ocean eddy current pixel classification, which consists of a convolutional encoder–decoder by a pixel-wise classification layer. The data in [30] origins from CMEMS’s sea surface height (SSH) maps. Yet its classification results are not so good as those of the statistic-based algorithms, which attributes to the application of the closed contour method. Dubbed DeepEddy [31], uses two principal component analysis (PCA) convolution layers to learn eddy features, and then implements a non-linear transformation through a binary hashing layer and block-wise histograms. It uses multi-scale features fusion for synthetic aperture radar (SAR) images. It has a similar performance as the statistic-based algorithms, but takes more computation consumption.

Our work follows the semantic segmentation branch and develops a PSPNet segmentation network for HFR data from the South China Sea. Different from the previous works, HFR data supports the fine-grained observation data for eddy detection. PSPNet implements eddy detection tasks in a convenient approach. It is a novel attempt to use HFR data to detect eddies on the offshore sea.

## 3. Data Preparation

Different from the previous detection work, all of our data were collected from observations by HFRs. In the experiment, two OSMAR-S radars [32] were depolyed at Shanliao and Xiaan in Fujian province of China to jointly measure the sea surface current field. The distance between them is about 60 km. The observed region is to the southwest of the Taiwan Strait, and the observation lasts for 80 days, from 11 January 2013, to 31 March 2013. OSMAR-S is a compact HFR system designed by Wuhan University, which uses monopole as the transmitting antenna and monopole cross-loop antenna as the receiving antenna. The center frequency of the transmitting signal is 13 MHz. The sweep bandwidth is 60 kHz and corresponding range resolution is 2.5 km. During the experiment, the data rate at each grid point is greater than 0.85, and the quality of the measured data is high. Figure 1 depicts the observed region and one current velocity field during the experiment.

In the observation, the sample rate is per 20 min. The number of all flow charts is 5760. Wherein, 5000 flow charts are processed as the training database, and the other 760 flow charts are used as the testing database. Following the previous work [29], we also adopt python-eddy-tracker software (PET14) [33] outputs as the training database for our eddy detection algorithm. To extract the most valuable information from the flow chart, the input image is set to 832 × 576 pixels. The characteristics of the 5000 flow charts include that there are an uncertain number of eddies distributed over all flow charts, the eddy size is relatively small, and the lifespan is about several hours. After being processed by PET14, the mask of the flow chart includes pixels of three categories: “0” means the background or no eddy data, “1” means the cyclonic eddy, and “2” means the anticyclonic eddy. The clustering shape of pixels marked as “1” or “2” is an arbitrary polygon. Figure 2 depicts an example of the pixel segmentation training couple.

## 4. Our Proposed Method

### 4.1. Architecture

The traditional semantic segmentation method uses FCN. Due to the characteristics of no requirements on image shape and the relatively high efficiency, FCN is popular in the general semantic segmentation task. However, for the eddy detection task, the decision of eddy boundary needs a preceding division. Meanwhile, the relationship between the eddy pixels needs to be reconsidered for space consistency. PSPNet [34] supports a global pyramid of pooling layers to handle additional contextual information. It fuses different-level features to integrate the semantic information and detail information, which matches the offshore eddy detection task. Therefore, the DEDNet architecture is based on the PSPNet architecture, which ensures accurate division and maintains spatial consistency.

The pipeline contains four steps: multi-layer feature extraction, pyramid pooling, concat, and predication. Figure 3 depicts the overall structure of DEDNet. In the multi-layer feature extraction, we adopt a pretrained deep residual network (ResNet-50) [35] to extract eddy features from the flow chart. ResNet-50 owes five convolution stages. The first is a convolution layer with 7 × 7 convolution kernels and two strides. The second includes 3 × 3 max pooling and three sequential stacked layers of 1 × 1 × 64, 3 × 3 × 64, and 1 × 1 × 256 convolution layers. The third includes four sequential stacked layers of 1 × 1 × 128, 3 × 3 × 128, and 1 × 1 × 512 convolution layers. The fourth includes six sequential stacked layers of 1 × 1 × 256, 3 × 3 × 256, and 1 × 1 × 1024 convolution layers. The fifth includes three sequential stacked layers of 1 × 1 × 512, 3 × 3 × 512, and 1 × 1 × 2048 convolution layers. The input size of the flow chart is 832 × 576, and the number of channels in the input layer is three. After the ResNet-50 model implements convolution for five times, the size of the feature map is transformed to 1/32 of the original flow chart, namely 26 × 18. Therefore, the processed flow chart experiences four sequential residual blocks to be transformed into a feature map, which contains abstract feature information of the flow chart. In the pyramid pooling, a pyramid pooling module is a hierarchical and global-optimum feature recombination module to connect feature information of different sizes in different regions. The module fuses four different pyramid features. As shown in Figure 3, the first red block of pyramid pooling represents the coarsest feature, which generates a single output through global pooling (1 × 1 bin). The latter three blocks of pyramid pooling divide the feature map into different subzones. Then, it adopts similar global pooling to each subzone and generates different bins with multi-level location information. Finally, the pyramid pooling module combines different (1 × 1, 2 × 2, 3 × 3, and 4 × 4) bins to represent pooling features of different sizes. There is a detail that should be noticed. If the pyramid contains N levels, a 1 × 1 convolution is added to each bin in order to ensure the global-feature weight, which can decrease the feature number to 1/N of the original feature number. In the concat step, different bins owe different low-dimensional feature maps, and then a bilinear interpolation is used to up-sample these feature maps to the uniform-size feature maps, which equals the original feature size. Finally, the feature maps on different levels, including the original feature map and the maps on the level of 1, 2, 3, and 4, are concatenated to achieve the pyramid pooling global feature. In the predication step, a convolution operator is implemented to generate the final prediction image.

From the opinion of pixel segmentation, eddy detection is deemed to be a multiclass classification problem. Different from the general segmentation tasks, eddy detection has the following two characteristics. The first is strong space correlations. The eddies are normally separated from each other. The offshore regions may present combinations of two different eddies. Considering the space consistency, DEDNet uses multi-size feature infusion to ensure the global optimum in space, which is suitable to handle special problems of strong correlation. The second is the accurate eddy boundary. The eddy boundary directly decides the sphere influence of an eddy, which needs to be precisely achieved for oceanographic applications, e.g., programming of the ship route. Meanwhile, in the flow chart, an eddy presents a shape of intertwined streamline, and its boundary is not constructed by a closed curve, which requires the model to have the capacity of distinguishing details. For this purpose, DEDNet uses four-level blocks of different sizes to recombine the abstract features and concatenates them for final predication.

The DEDNet adopts K-fold cross-validation (K = 10) as the training strategy and stochastic gradient descent (SGD) as the optimizing strategy. The 5000 flow charts are divided into 10 sets with each set containing 500 flow charts. On each run, nine sets are regarded as training datasets, and the remaining set are regarded as the validation dataset. Accuracy can be achieved by such dataset combinations for DEDNet training. The experiment runs continuously until every set has been utilized as the validation dataset. The computed average accuracy is identified as the final accuracy.

### 4.2. Loss Metric

In multiclass classification task, the categorical cross-entropy cost function is normally used to evaluate the performance of the model. However, overlap-based metrics are more suitable for pixel segmentation tasks, because the overlap-based metrics directly compute the probability of a pixel belonging to a specific category, and the computation approach is more direct. The dice coefficient [36] is a popular and widely-used cost function in such segmentation tasks. It can reduce the possibility of over fitting. It is also a direct evaluation index to represent the neural network’s performance. Assuming that $p_{i}$ is the i-th pixel of predicted image *P* and $q_{i}$ is the i-th pixel of ground truth image *Q*, the dice coefficient can be computed as
$$DiceCoef\left(P,Q\right)=\frac{2\sum_{i}p_{i}\times q_{i}}{\sum_{i}p_{i}+\sum_{i}q_{i}}$$
where the numerator is the twice of the intersection area between *P* and *Q*; the denominator is the sum of the areas of *P* and *Q*.

## 5. Experiment

### 5.1. Experiment Setup

The experiment is implemented on the Pytorch framework with a Tensorflow backend. The hardware includes a NVidia GTX 1060 GPU card with a 6 GB memory and an Intel I5-6600 K CPU at 3.5 GHz. The experiment adopts N-fold cross validation to prevent over fit. The N is set as 10. In the training process, 5000 flow charts are divided into 10 groups with each group having 500 training images. One group is randomly sorted as the validation dataset, and the other nine groups are used for training. The experiment runs 10 times, until every group has been used for validation. The training accuracy is the average accuracy of all the experiments.

### 5.2. Performance Assessment

We also present an FCN-based eddy detection network to compare with our DEDNet. Differently from the convolutional neural network (CNN), the fully convolutional network transforms the last three full connection layers to three convolution layers. The FCN-based network contains eight convolution layers and one pixel-wise prediction, which has the capacity to input an image of arbitrary size. The input size of the flow chart is 832 × 576, and the number of channels in the input layer is three. The image passes eight convolution layers to generate a feature map, which is processed by an up-sampling operator to recover the original size. After this approach, unlabeled feature samples achieve pixel-level classification with the original spatial information. In a word, DEDNet is a deep eddy detection neural network with PSPNet skeleton on HFR data.

Figure 4 reveals the eddy identification comparison among the ground truth, a FCN-based detection network, and DEDNet. Here the segmentation result by PET-14 is regarded as the ground truth. The ground truth algorithm uses streamline computing to detect eddies and distinguishes the boundary depending on the streamline length and twisting degree. It can be concluded that DEDNet gives more accurate eddy locations and boundaries than the FCN-based network. Meanwhile, there is a gap between both DEDNet and FCN-based networks and the ground truth algorithm. DEDNet and FCN-based networks determine the boundaries depending on the pixel classification. Even though the boundary pixels are consistency, the boundary cannot be a perfect curve while fitting with a closed streamline, which attributes to the imperfectness of pixel locations. As a conclusion, DEDNet performs better than FCN-based network, but performs worse than the ground truth algorithm. Considering that the annotated flow charts are generated by the ground truth algorithm, DEDNet’s actual performance is limited by the annotated flow charts.

Table 1 compares the results of dice coefficient, Intersection over Union (IOU), and predication accuracy obtained by DEDNet and the FCN-based network. As a whole, the predication accuracy by DEDNet is 3.72% higher than that by the FCN-based network. DEDNet also performs better than the FCN-based network in terms of both the average dice coefficient and IOU. The improvement is 0.052 and 0.081, respectively. DEDNet has also a smaller value of standard deviation, which implies that DEDNet is more stable than the the FCN-based network. In detail, the dice coefficient in case of none eddy is the highest, which means pixels under this situation are easier to distinguish. The pixels of anticyclones and cyclones owe similar distinguishing difficulty, corresponding to their similar dice coefficients. Figure 5 depicts the eddy detection results of three categories. In DEDNet, it is found that the error rate mainly originates from the errors between eddy and none eddy, there are relatively low errors between anticyclones and cyclones. In the FCN-based network, it is found that the fuzzy decision between anticyclones and cyclones is relatively severe.

### 5.3. Detection Analysis

Figure 6 depicts the histogram of the eddy radius by DEDNet, the FCN-based network and the ground truth. It can be concluded that most radii of the offshore eddies are from 6 to 15 km. The number of eddies with a radius greater than 24 km is relatively small. This is mainly because the water in the offshore region is generally shallow, which makes it difficult to generate large-size eddies. The short duration of these small-size eddies may be due to the comprehensive topography and weather in the Taiwan strait. As can be seen from Figure 6, the number of eddies detected by DEDNet is similar to the ground truth results, but the FCN-based network shows an obvious difference. This also demonstrates that DEDNet performs better than the FCN-based network.

Figure 7 depicts the numbers of anticyclones, cyclones, and none eddies detected by the ground truth, FCN-based eddy network, and DEDNet. The numbers of the anticyclones and cyclones are similar, say about 1800, and that of the none eddies are about 1300. The statistical data imply that the offshore submesoscale eddies occur in the Taiwan strait with a relatively high frequency. The decision by the ground truth algorithm is stricter, which requires that the eddies’ twist degrees exceed a specific threshold and the eddy shapes are complete. Therefore, the difference between the network-based detection method and the ground truth algorithm mainly originates from two cases. The first is that several streamlines surround each other in the flow chart, but the twist degree does not reach the threshold. The network-based eddy detection needs more reasonable data to approach the threshold. Therefore, part of no eddies is mistaken as eddies. The second case is that some eddies are just located near the boundary of the observation region. Only part of these eddies can be seen in the flow chart, while other parts are out of sight. Due to the incomplete shapes, they are not able to be detected as eddies by the ground truth algorithm, and are counted into the category of no eddy. However, the network-based eddy detector can regard them as eddies due to the abstract eddy features. The network-based detection may miss a small number of eddies, but it can recognize a few more incomplete eddies. Therefore, the network interprets more eddies than the ground truth.

## 6. Conclusions

This paper investigates the new deep learning technology for eddy detection tasks. Differently from the previous work based on the SSH data and the objective detection skeleton, we propose a deep eddy detection neural network with the pixel segmentation skeleton on HFR data, namely DEDNet. Firstly, we construct an offshore eddy detection dataset, which originates from the true measurement data by two HFR systems in the South China Sea. Secondly, based on the PSPNet skeleton, a spatial global optimum and strong detail-distinguishing pixel segmentation network is constructed to predicate eddy distribution in the flow charts. Thirdly, an FCN-based eddy detection network is also presented for eddy detection on HFR data for comparison. The experiments show that DEDNet’s predication accuracy, average dice coefficient, and IOU are, respectively, improved by 3.72%, 0.052%, and 0.081% more than the FCN-based network. It can be concluded that DEDNet performs better than the FCN-based eddy detection network and thus has potential of approaching the classical statistics-based methods.

In the future, we will attempt to explore, but not limited to the following studies. The first one is the semi-supervised deep eddy detection, which is urgent for this situation. Due to the high cost of eddy labeling and flow chart collection, the number of labeled flow charts is relatively small. The critical problem is how to improve the eddy detection accuracy with a small number of well-labeled images and a huge number of unlabeled images. The second is to construct unique eddy benchmark datasets and an eddy testing platform, since there is no benchmark based on true observation data for eddy detection so far. The third one is to collect eddy data in a larger offshore region and monitor the evolution of multiple eddies. Multiple-label and longer-term training datasets need to be constructed to support accurate detection of eddies. Meanwhile, an enlarged observation region will also be considered to explore more eddies with complete shapes.

## Figures and Tables

**Figure 1 sensors-21-00126-f001:**
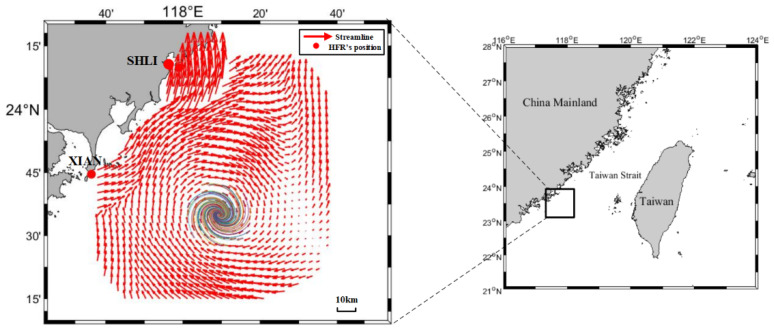
The observed region in Taiwan Strait.

**Figure 2 sensors-21-00126-f002:**
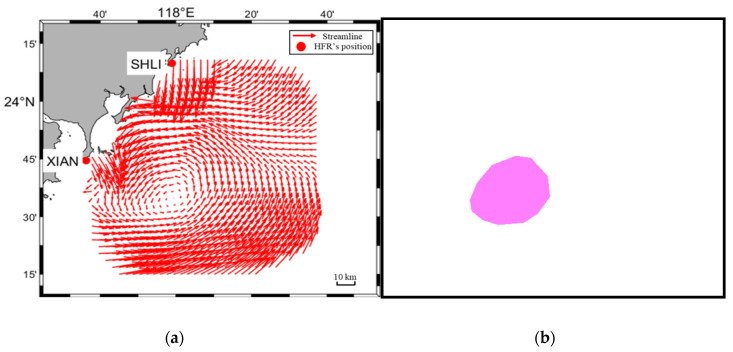
Example of pixel segmentation training couple. (**a**) Original flow chart; (**b**) Eddy segmentation mask results.

**Figure 3 sensors-21-00126-f003:**
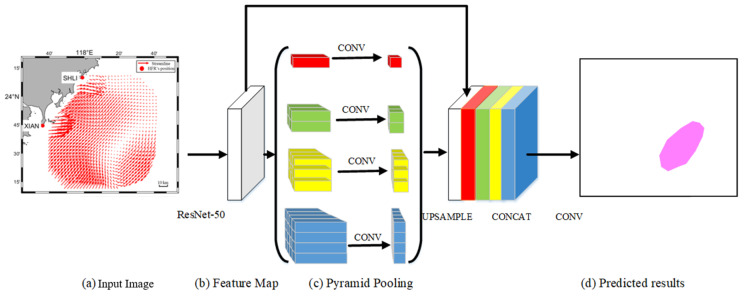
The DEDNet’s structure.

**Figure 4 sensors-21-00126-f004:**
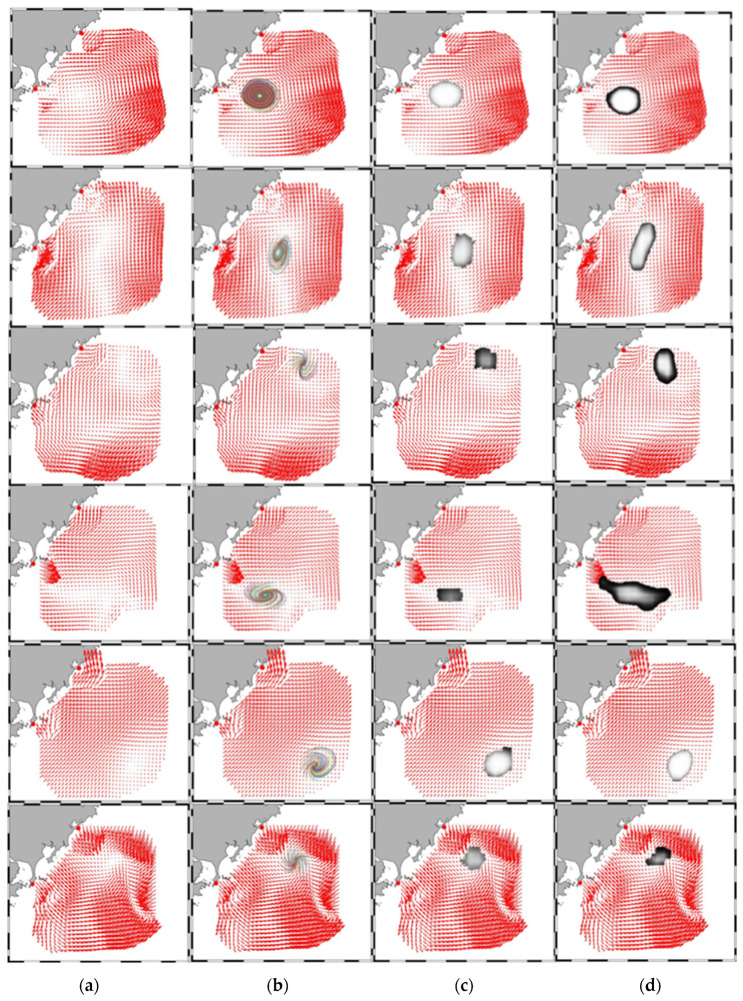
Comparison of eddy detection results between ground truth, FCN-based eddy network and DEDNet. (**a**) Original flow chart; (**b**) Ground truth eddies; (**c**) Eddy detected by DEDNet; (**d**) Eddy detected by a FCN-based network.

**Figure 5 sensors-21-00126-f005:**
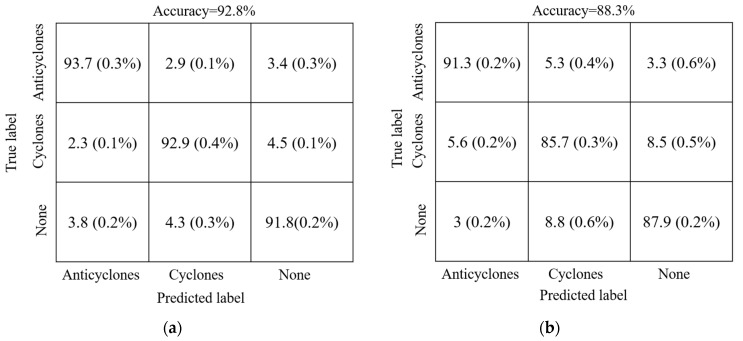
The eddy detection results of three categories. (**a**) DEDNet; (**b**) FCN-based network.

**Figure 6 sensors-21-00126-f006:**
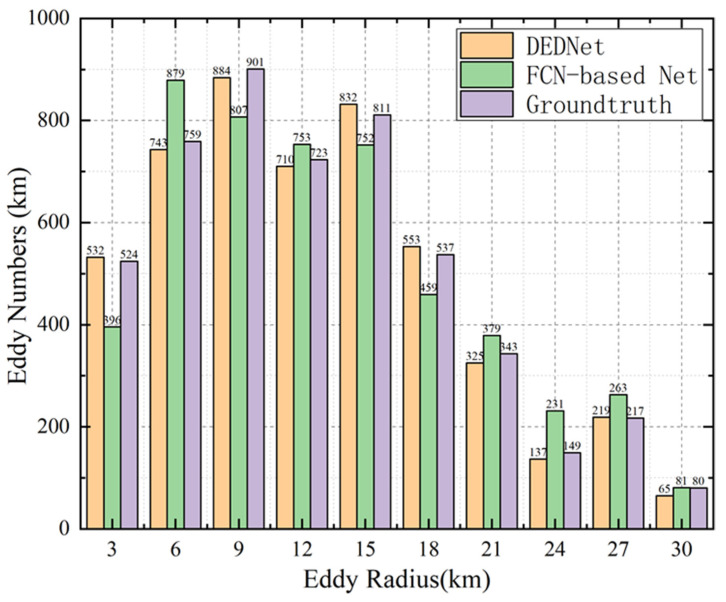
The eddy radius distribution detected by ground truth, FCN-based eddy network, and DEDNet.

**Figure 7 sensors-21-00126-f007:**
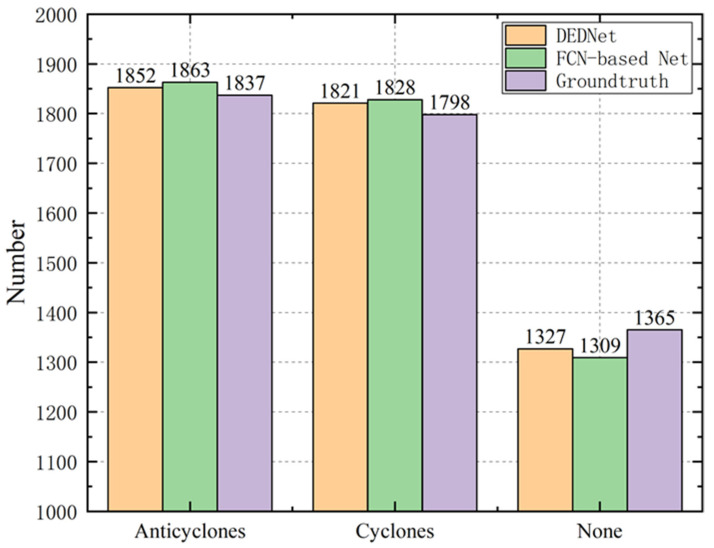
The different eddy types detected by the ground truth, FCN-based eddy network, and DEDNet.

**Table 1 sensors-21-00126-t001:** The dice coefficient, IOU, and predication accuracy obtained by DEDNet and FCN-based Net on 30-times experiments.

Algorithm	Dice Coefficient			Average Dice	IOU	Predication
	Anticyclones	Cyclones	None	Coefficient		Accuracy
DEDNet	0.873 (0.002)	0.861 (0.001)	0.936 (0.001)	0.89 (0.001)	0.802 (0.001)	91.38% (0.08%)
FCN-based Net	0.795 (0.002)	0.824 (0.003)	0.897 (0.001)	0.838 (0.002)	0.721 (0.002)	87.66% (0.11%)

## Data Availability

The data presented in this study are available on request from the corresponding author. The data are not publicly available due to privacy.

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
