# Peer review of "DEDNet: Offshore Eddy Detection and Location with HF Radar by Deep Learning"

_sensors, 2020, doi:10.3390/s21010126_

Round 1

Reviewer 2 Report

Original Submission

Recommendation

Rejection

Comments to Author:

Title:  

 DEDNet: Offshore Eddy Detection and Location with HF Radar by Deep Learning.

Overview and general recommendation.

This paper is demonstrating an approach to help to detection of Eddy current. These kinds of data are always welcomed to MDPI journals including RS, Sensors, Imaging etc. However, I think that this paper is better fit to the RS_MDPI journal. The work is OK, and lots of good works done here. But we do not see any application here. English is very poor and needs to be improved. I think paper is very chaotic and really must be written again. Abstract is really bad; not reflecting the material. Introduction is saying noting, but a bit about the DL. Honestly, I like the work: very good and efficient, but the writing is a disaster. Also, the material is not sufficing: Much experiments, and reliable comparisons must be presented.

Detailed comments:

lines 53. … of physical > I think “on” would be better.

lines 57. The eddy center is thus located? Explain. Not good.

lines 65. Put a ref here.

lines 74. Put a ref here.

lines 93. … a anchored?

lines 96-97. Rephrase please.

lines 102. …an more?

lines 121-122. Redundancy.

Fig.1 and 2 and 3… NO legend; no Scale? Even No location map.

Fig 3 caption is 6 lines after the image.

And lots more...

Reviewer 3 Report

Detailed comments:

Line 4: Please update the sentence for punctuation.

Line 22: Please check and update the keywords for capitalization.

Line 29 and 30: Please check the sentence for punctuation. The usage is X, Y, and Z. the second "," is missed here. 

Line 40: Please change "with" to "at". The "at" seems to be more appropriate.

Line 50: Please update the formatting for the in-text citations. This is a typical comment, therefore, double-check the rest of the document for this issue. For example lines 55, 63, 

Line 51: Same comment as Lines 29 and 30. 

Lines 57 to 60: Please rewrite the sentence as it is hard to follow.

Lines 66 and 67: the VG should come immediately after vector geometry, not after the algorithm.  

Line 70: Please update the sentence for clarity. Do you mean object detection?

Lines 89 and 90: Please avoid paragraphs with less than 3 sentences. 

Lines 97 and 110: Please update the sentence for the usage of "e.g.," and "i.e.,".

Line 104: Same as Lines 29 and 30.

Line 108: Same as Lines 29 and 30.

Line 110: Please provide references for the Okubo-Weiss algorithm.

Line 115: Please provide a reference for VG and WA algorithms. 

Line 120: Please provide a reference for the geometry-based algorithm requirement and its efficiency. 

Lines 121 and 122: Please update the sentence for clarity. 

Line 126: A "," is missed before "and".

Lines 127, 130, and 132: Please update the formatting for the in-text citations.

Lines 147 and 148: Please avoid 2 sentence paragraphs. Paragraphs should be 3 sentences or more.

Line 155: Please provide references for the mentioned methods developed by Shanliao et al. 

Line 168: Please provide a reference for the software used, the version used, potential parameters used to prepare the training data. 

Section 4: This section requires a lot of revisions and improvements. Please discuss the input size and number of channels used in the input layer. Then, discuss each part of the network in detail, including the architecture and a reference to the model (e.g., ResNet-50). Here are a few main comments that must be addressed:

1- Firstly, many details and references are missed in regard to the network. 

2- Figure 3 caption is not shown in the correct location.

3- The details in regard to ResNet-50 is not provided.

4- The tags in the figure seems to be inaccurate. For example (a)flow chart?

5- Did you use skip connection from ResNet to Concat layer?

6- Please discuss the training strategy used to develop the model. Did you use gradient descent?

7- Could you please provide more detail about the Dice coefficient including a reference?

Line 240: Please discuss the FCN-based model in more detail similar to the model in section 4.

258: Why IOU is only reported as the performance measure. It is beneficial to know the precision and recall values in addition to IOU. Also, can the authors compare the models using other techniques not just based on accuracy? 

Table 1 is separated into two segments. Please update the spacing to avoid this issue. 

Round 2

Reviewer 2 Report

The authors have been responded almost all of the comments, and seems to me they are in good understating of the material; the paper is improved remarkably, and in this form, I am positive to accept it,

Good Luck!